# Objective Methods of 5-Aminolevulinic Acid-Based Endoscopic Photodynamic Diagnosis Using Artificial Intelligence for Identification of Gastric Tumors

**DOI:** 10.3390/jcm11113030

**Published:** 2022-05-27

**Authors:** Taro Yamashita, Hiroki Kurumi, Masashi Fujii, Takuki Sakaguchi, Takeshi Hashimoto, Hidehito Kinoshita, Tsutomu Kanda, Takumi Onoyama, Yuichiro Ikebuchi, Akira Yoshida, Koichiro Kawaguchi, Kazuo Yashima, Hajime Isomoto

**Affiliations:** 1Division of Gastroenterology and Nephrology, Faculty of Medicine, Tottori University, 36-1, Nishi-cho, Yonago 683-8504, Japan; kurumi_1022_1107@yahoo.co.jp (H.K.); fujii-masashi@jptecman.com (M.F.); sakaguchitakuki@yahoo.co.jp (T.S.); t-hashimoto.1022@tottori-u.ac.jp (T.H.); tsutomu.kanda.s@gmail.com (T.K.); golf4to@yahoo.co.jp (T.O.); ikebu@tottori-u.ac.jp (Y.I.); akirayoshida1021@yahoo.co.jp (A.Y.); koichiro@tottori-u.ac.jp (K.K.); yashima@tottori-u.ac.jp (K.Y.); isomoto@tottori-u.ac.jp (H.I.); 2Division of Gastroenterology, Sanin Rosai Hospital, 1-8-1, Kaike Shinden, Yonago 683-0002, Japan; hide.19871207@gmail.com

**Keywords:** photodynamic diagnosis, 5-aminolevulinic acid, LAB color space, neural network

## Abstract

Positive diagnoses of gastric tumors from photodynamic diagnosis (PDD) images after the administration of 5-aminolevulinic acid are subjectively identified by expert endoscopists. Objective methods of tumor identification are needed to reduce potential misidentifications. We developed two methods to identify gastric tumors from PDD images. Method one was applied to segmented regions in the PDD endoscopic image to determine the region in LAB color space to be attributed to tumors using a multi-layer neural network. Method two aimed to diagnose tumors and determine regions in the PDD endoscopic image attributed to tumors using the convoluted neural network method. The efficiencies of diagnosing tumors were 77.8% (7/9) and 93.3% (14/15) for method one and method two, respectively. The efficiencies of determining tumor region defined as the ratio of the area were 35.7% (0.0–78.0) and 48.5% (3.0–89.1) for method one and method two, respectively. False-positive rates defined as the ratio of the area were 0.3% (0.0–2.0) and 3.8% (0.0–17.4) for method one and method two, respectively. Objective methods of determining tumor region in 5-aminolevulinic acid-based endoscopic PDD were developed by identifying regions in LAB color space attributed to tumors or by applying a method of convoluted neural network.

## 1. Introduction

Recent developments in image enhancement technologies have greatly improved the endoscopic diagnosis of gastric tumors. Narrow band imaging (NBI) with the aid of magnification is able to better visualize blood vessel contraction in the mucosa [1,2,3].

However, the first endoscopic identification of gastric tumors is usually performed by white-light endoscopy, because magnification is needed for full utilization of the NBI method in the identification of gastric tumors. Photodynamic diagnosis (PDD) endoscopy after oral intake of 5-aminolevulinic acid (5-ALA) has proven useful for detecting gastric tumors [4,5,6]. A recent study found additional lesions that were identified as tumors in 13 cases with the use of PDD endoscopy [7]. This indicates that there is a need to reduce the risk of missed diagnosis of gastric tumors by improving the PDD endoscopy method. Classification of a defined region as PDD positive or negative is determined by an expert endoscopist [7]. A more objective method of classification would be preferable as this would reduce the likelihood of tumors being overlooked. Protoporphyrin IX is the main fluorescent metabolite of 5-ALA in the living cell. It has primary peak fluorescence at 630 nm and a fluorescence spectrum ranging from 620 to 700 nm [8]. Upon exposure to ultraviolet (UV) light, the tumor region appears red with PDD due to increased fluorescence light from protoporphyrin IX (PPIX) in tumor cells compared to that in healthy cells.

Therefore, it could be possible to differentiate tumor and non-tumor cells by objectively defining the color of a segmented region in the PDD image. We have chosen LAB color space for that purpose. LAB color space corresponds to the human perception of color, where ‘l’ represents brightness and ‘a’ and ‘b’ represent color tone. In this study, we have used the multi-layer neural network method to determine the region assigned to the tumor in LAB color space.

Another strategy is applying the Convoluted Neural Network (CNN) method. CNN has been recently employed for the identification of specific objects in various situations such as autonomous driving [9]. Applying artificial intelligence (AI), especially the CNN method, to white-light or NBI endoscopy has been recently investigated [10], where AI applications can have a more accurate performance in detecting abnormality than endoscopists. With the CNN method, not only identifying objects but also segmenting images into classified objects can be performed [8]. Performing segmentation according to the estimated histology with the CNN method has recently been investigated for colonic endoscopy [11,12]. For gastric endoscopy, we have not seen any systematic study of performing segmentation with artificial intelligence. In this study, we have used the CNN method to diagnose gastric tumors and to differentiate tumor and non-tumor regions in PDD images.

One feature which has an impact on PDD efficiency is the photobleaching effect, where the fluorescence light degrades as the PPIX is exposed to light [8,13]. We have investigated how photobleaching affects the effectiveness of the developed methods.

## 2. Materials and Methods

### 2.1. Endoscopic Systems

We used a prototype PDD endoscopic system (Fuji Film Medical Co., Tokyo, Japan) [7,14]. The system has white light, NBI, and PDD inspection modes. When in PDD mode, UV light with a center frequency of 410 nm is either laser [7,14] (denoted as laser-based system) or Light Emitting Diode (LED) (denoted as LED-based system). Fluorescence is detected by the complementary metal oxide semiconductor (CMOS) detector of the endoscope.

### 2.2. Endoscopic Procedures

PDD screening was performed on the day of endoscopic submucosal dissection (ESD), except for two lesions (#20 and #24) of case #20, which were surgically resected. Patients were administered 5-ALA orally with a dose of 20mg/kg, from 2 to 4 h before the diagnostic procedure. For each lesion identified as a tumor, several PDD images were taken at a midrange distance without magnification.

ESD was performed after the PDD screening, according to the standard procedure [15]: marking, injecting liquid composed of hyaluronic acid and saline into the submucosa, making an incision and dissecting with submucosa level. Light shading of 24 h after 5-ALA intake was implemented to avoid any photosensitivity [7].

### 2.3. Lesions in the Analysis

27 lesions in 20 patients (age from 58 to 78 (median 73), two are female) of differentiated adenocarcinoma and adenoma were included in this study. One lesion with poorly differentiated histology was excluded from this study as we previously reported that the efficiency was low with the 5-ALA-based PDD for poorly differentiated adenocarcinoma of the stomach [14].

### 2.4. Determination of Tumor Region

The tumor region of each PDD image was determined with reference to the pathological result of the resected specimen (ESD or surgical operation), together with white light and blue light image (BLI) endoscopic pictures. Tumor regions and non-tumor regions were annotated for each PDD image. In both methods described below, area efficiency and area 1-specificity for PDD images were calculated from the determined region as follows:

Area efficiency = fraction of the area determined as tumor by the methods/the area determined as a tumor with reference to the pathological results

Area 1-specificity = fraction of the area determined as tumor by the methods/the area determined as non-tumor with reference to the pathological results.

### 2.5. Diagnosis of Tumors

For the majority of the lesions (22 in 27 lesions), several PDD images taken for each lesion were used for learning and validation, so as to maximize the learning power and estimate the uncertainty of the methods. Comparison of the results of the PDD to the diagnosis by the expert endoscopists was also performed.

The positivity of the lesion was defined as follows (i.e., diagnosed as a tumor): if at least one PDD picture had an area efficiency >10% as defined by the respective method.

The methods were implemented in Microsoft Visual Studio (Microsoft, Redmond, WA, USA) with the programming language Python 3.

### 2.6. Method One: Identification of Tumor Image Using Lab Color Space and a Multi-Layer Neural Network

Here, we performed PDD of 9 lesions in 6 cases using the LED-based system. We examined 9 lesions. Due to the limited number of cases, we did not have separate training and validation lesions. We used a neural network of three layers with an input of LAB values of each pixel of the PDD image and an output of two classes: tumor and non-tumor. The average brightness of each image was normalized to the same value. Parameters of the learning procedure, for example, node size of each layer of neural network (=300, 200, and 100, for the first, second and third layer, respectively) or activation function (=logistic) were tuned or selected, such that the learning converged faster with smaller error.

### 2.7. Method Two: Diagnosing and Determining Tumor Region in PDD Images Using Convoluted Neural Network and Semantic Segmentation

We evaluated 27 lesions in 20 cases with pathologies of well or moderately differentiated carcinoma or adenoma. A CNN was applied to the PDD images. Semantic segmentations were performed for each image and classified into three classes: tumor, non-tumor (mucosa other than tumor), and background (anything other than tumor or non-tumor, for example, endoscopic shaft or the region which is too dark for evaluation).

A data augmentation method was applied to the training pictures by rotating, magnifying, and flipping. As the RGB gains of the color setting were not as sensitive as in method one, we divided the 27 lesions (18 lesions by the laser-based system, 9 lesions by the LED-based system) into two groups: 12 lesions for training and 15 lesions for validation. Lesions were divided such that both adenoma and adenocarcinoma lesions were included in training and validation. Number of epochs for CNN training, which was the number of multiple usage of data sets, was selected as 120, considering that the error calculated for validated lesions stopped improving at around 120 epochs.

## 3. Results

The longest diameter of the lesion included in the analysis was from 5 to 40 mm (median 18 mm), and the estimated area, assuming the lesion was the ellipse, was from 11.8 mm^2^ to 1068.1 mm^2^ (median 181.4 mm^2^). Table 1(a–c) show the summary of the morphology, histological diagnosis, and histological depth, respectively, of the 27 lesions included in the analysis. Further detailed features of the lesions are shown in Appendix A.

Figure 1(a1–c1,a2–c2) show examples of the white light and PDD images together with the annotated images for lesion #19 and #20.

### 3.1. Method One

The areas in PDD images with corresponding values in LAB color space for tumor points and non-tumor points are shown in Figure 2. Points in LAB color space assigned as tumors by method one are overlaid and shown as the red-smoked area. Tumor and non-tumor points are separated in LAB color space such that the tumor points have a higher “a” value.

Figure 3(a1,a2) show examples of the PDD image (lesion #19 and #20, respectively). PDD images were overlaid by images with tumor regions determined by method one, which are depicted in red. The efficiency of diagnosing tumors was 77.8% (7/9, where the two lesions (lesion 22 and 24) are not detected as tumors by method one). The area efficiency was 35.7% (0.0–78.0). The area 1-specificity was 0.3% (0.0–2.0).

Figure 4 shows the calculated area efficiency (a) and the 1-specificity (b) when method one was applied to nine lesions. Lesions #8, #22, and #24 had low efficiency, while lesion #24 had zero efficiency. The three lesions (#8, #22) were diagnosed as weak positives by the expert, and lesion #24 was diagnosed as negative by the expert. The remaining seven lesions were diagnosed as strong positives, showing the consistency of method one with the diagnosis by the expert endoscopists.

### 3.2. Method Two

Figure 5(a1,a2) shows examples of the PDD image (lesion #19 and #20, respectively); PDD image with segmented image overlaid (b) in which tumor region determined by method two is depicted in brown, and non-tumor region is depicted in green. The tumor regions are well segmented by method two.

Figure 6 shows the calculated area efficiency (a) and area 1-specificity (b) when method two was applied to 12 lesions used for the training.

Figure 7 shows the calculated area efficiency (a) and area 1-specificity (b) after the application of method two to 15 lesions used for validation. The efficiency of diagnosing tumors for validated lesions was 93.3% (14/15, where lesion #24 was not detected as a tumor by method two). The area efficiency was 48.5% (3.0–89.1). The area 1-specificity was 3.8% (0.0–17.4). The efficiency and 1-specificity presented here are for validation lesions. Lesions #18, #22, #24, #25 and #27 had relatively low area efficiencies (<20%). Lesions #22, and #24 (taken by the LED-based system and examined by method one) also had low efficiency by method one, which showed overall consistency between method one and method two.

### 3.3. PDD Endoscopy Identification of Previously Undetected Lesions

Among the 27 lesions considered here, three lesions (lesions #3, #17, and #21) were detected as tumors for the first time on PDD endoscopy. Figure 8(a1,a2) shows the PDD images of lesion #17, and (c1) and (c2) show the results with method two applied (the lesion was used for validation by method two). As is shown, the lesion was detected as a tumor by method two. The lesions were biopsied and diagnosed pathologically and were later resected by ESD. These examples show the effectiveness of not only PDD endoscopy but also of the objective methods developed here to reduce missed diagnoses.

### 3.4. The Effect of Photobleaching

The color of the PDD picture changed as the lesion was exposed to the endoscopic light due to the degradation phenomenon of photobleaching [8,13]. The effect of photobleaching on the methods developed here was examined by applying each method to PDD images of lesion #12 (adenoma). Images were sequentially taken with exposure to the examining light of the PDD endoscopy at observation times of 0, 30, 60, 180, and 600 s (Figure 9). As shown, the red color of the PDD images became less intense, and the area identified as a tumor by our methods decreased as the observation time increased.

Figure 10a shows the points of PDD image in LAB color space for tumor and non-tumor regions of lesion #12. Figure 10b shows the efficiency of methods one and two as a function of observation time length from 0 to 600 s. As shown, the efficiencies decreased as a function of observation time for both methods. The effect was more prominent for method one, where the area efficiency decreased quickly as the observation time increased. For method two, the area efficiency decreased from 1 to 0.6 with the observation time of 180 s. In method two, which used the CNN algorithm, the color difference between the specified region and the adjacent region was taken into account for the determination of the tumor region. In contrast, method one used absolute values of l, a, and b in LAB color space. The color of both tumor and non-tumor regions was affected by photobleaching and appeared less red in color (with low a value). We, therefore, assumed that method two had an advantage over method one with regard to the photobleaching effect.

## 4. Discussion

Herein, we developed two methods for the detection of gastric tumors in PDD images, using artificial intelligence. Method one uses a multi-layer neural network to identify the color of tumors in LAB color space, while method two uses a CNN method to detect tumors and determine their region. The two methods are consistent, such that the high- and low-efficiency lesions identified by method one are also high- and low- efficiency by method two. The objective methods developed were validated by comparing these to those of the pathologic examination.

Moreover, our methods consistently identified the same lesions defined as strong-positive, weak-positive, and negative as identified by the expert endoscopist. This is the first report applying AI to PDD endoscopy to improve objectivity. Method two was found to be less sensitive to the photobleaching effect than method one due to the use of CNN which can identify the difference in color between adjacent regions.

Next, we examined features of low area efficiency lesions. Some cases (lesions #4, #18, #24, and #27) with low area efficiencies have pathological features in which the non-tumor glands are partially present on the surface of the lesions. The example of such cases is shown in Appendix A (lesion #4) and Appendix A (lesion #18). Lesion #24 also had low area efficiency as determined by method two or zero efficiency by method one. This lesion was an EBV-positive gastric tumor, which is one of the special types of gastric tumor [16] whose pathology has distinct features of prominent tumor-infiltrating lymphocytes around the tumor glands, which was also noticed for lesion #24. This feature may account for low fluorescence as seen by the 5-ALA-based PDD method as tumor-infiltrating lymphocytes on the tumor surface reduce the amount of UV light reaching the tumor glands.

Another notable fact is that two lesions (lesions #17 and #21) that were first detected by expert diagnosis by means of PDD were positive with method two, which shows the effectiveness of the objective method we have developed as a tumor-detection method. The result shows the possible efficacy of the method to reduce potential misidentifications, which can be applied after or during the endoscopic procedure.

Furthermore, the area efficiency and specificity of the methods show good performance in most cases, showing that the methods could be applied for identifying the tumor region. As far as we know, this is the first report applying AI to perform segmentation into classified estimated histology (tumor) for gastric endoscopy, which becomes possible with the aid of PDD. Larger studies with more samples are needed for further validation of the efficacy of our methods.

With the achieved objectivity and ability, the developed methods could make PDD endoscopy to be valuable for general screening of the upper gastrointestinal tract. Currently, the application is limited to the occasion of detailed inspection, due to the cost of 5-ALA (~600 dollars per one PDD endoscopy) and the need for hospitalization for 24-h light shading.

## 5. Limitations

This study was confined to the diagnosis of differentiated gastric tumors (adenoma and adenocarcinoma), excluding poorly differentiated adenocarcinoma which was reported to have low efficiency in PDD endoscopy. It is suggested that the non-tumor glands at the surface of the tumor can lower the efficiency of the segmentation of the region, and the diagnosis itself. In this regard, we have not performed PDD for submucosal gastric tumor which is supposed to have low efficiency because of the non-tumor glands at the surface. Due to limited case numbers, the specificity for detecting other kinds of non-malignant lesions, for example, gastric ulcer, scar, hyperplastic polyp, and fundic gland, is not known and will require further investigation.

The photobleaching effect may become a source of uncertainty, as has been mentioned earlier, which could limit the observation time of PDD.

Further limitations include the annotation uncertainty of the tumor region in the PDD images. The annotation is performed by comparing the image to the pathological segmented results assessed by the pathologist; procedural uncertainty mainly arises from determining the position of the PDD images compared to the pathological results.

The procedures were developed on the Python software platform and took from several to ten seconds per one image executing on a personal computer. For real-time application during the endoscopic examination, the processing time should be shortened, which is possible by executing on C++ software platform or with the specialized system of small latency, for example using the real-time processor.

## 6. Conclusions

Objective methods of determining tumor regions in 5-ALA-based endoscopic PDD were developed by identifying regions in LAB color space attributed to the tumor by using a method based on CNN. Our proposed methods are useful not only for identifying the gastric tumor but also for determining its region.

## Figures and Tables

**Figure 1 jcm-11-03030-f001:**
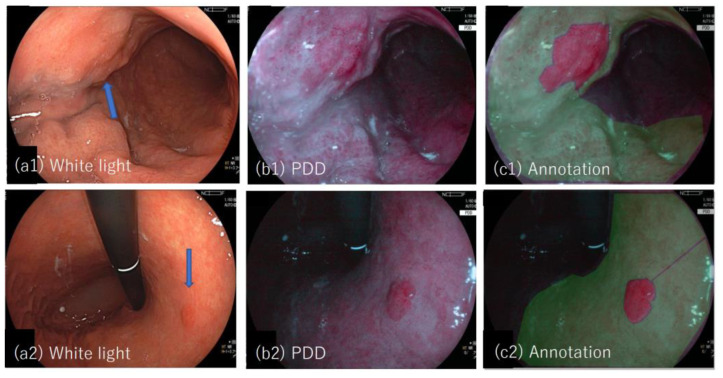
Tumor lesions examined by LED-based system, (**a1**–**c1**) show lesion #19 (adenocarcinoma) (**a1**–**c1**) and lesion #20 (adenocarcinoma); (**a1**,**a2**) White light images where tumor locations are indicated by blue arrows; (**b1**,**b2**) photodynamic diagnosis images; (**c1**,**c2**) photodynamic diagnosis images with an annotated image overlaid, in which tumor region determined with reference to the pathology of the specimen is depicted in red and non-tumor region is depicted in green.

**Figure 2 jcm-11-03030-f002:**
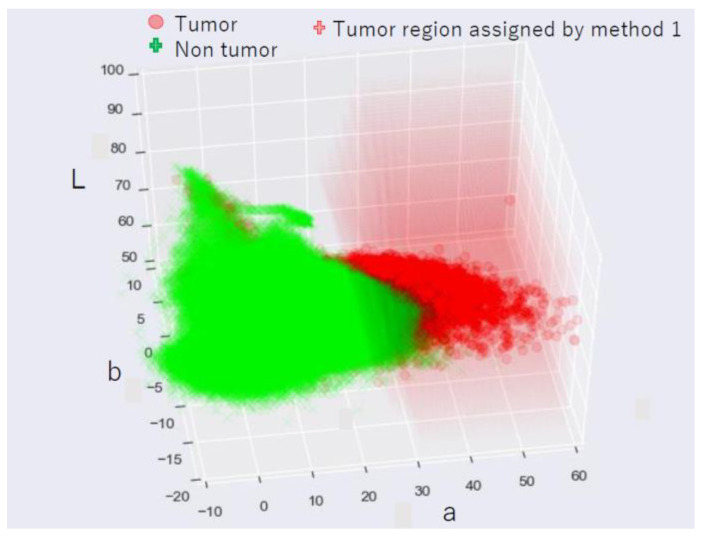
Tumor and non-tumor in the LAB space and the area assigned to tumor by method one: Points in the photodynamic diagnosis images are plotted with corresponding values in LAB color space, for tumor points (red circle) and non-tumor points (green cross). Points in LAB color space classified as tumors by method one are plotted as red-smoked areas.

**Figure 3 jcm-11-03030-f003:**
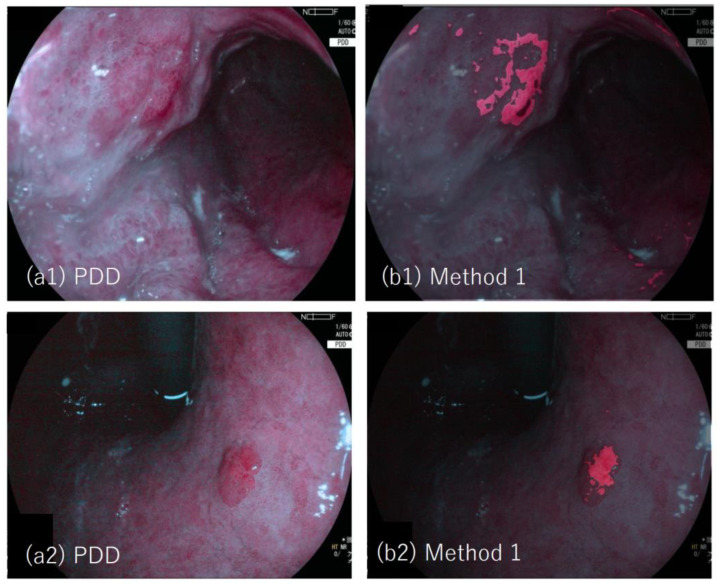
Example of PDD images and the results applied by method one: (**a1**) Photodynamic diagnosis (PDD) image of lesion #19 (adenocarcinoma), (**b1**) PDD image with an annotated image overlaid, in which tumor region determined by method one is depicted in red; (**a2**) PDD image of lesion #20 (adenocarcinoma); (**b2**) The PDD image with an annotated image overlaid, in which tumor region determined by method one is depicted in red.

**Figure 4 jcm-11-03030-f004:**
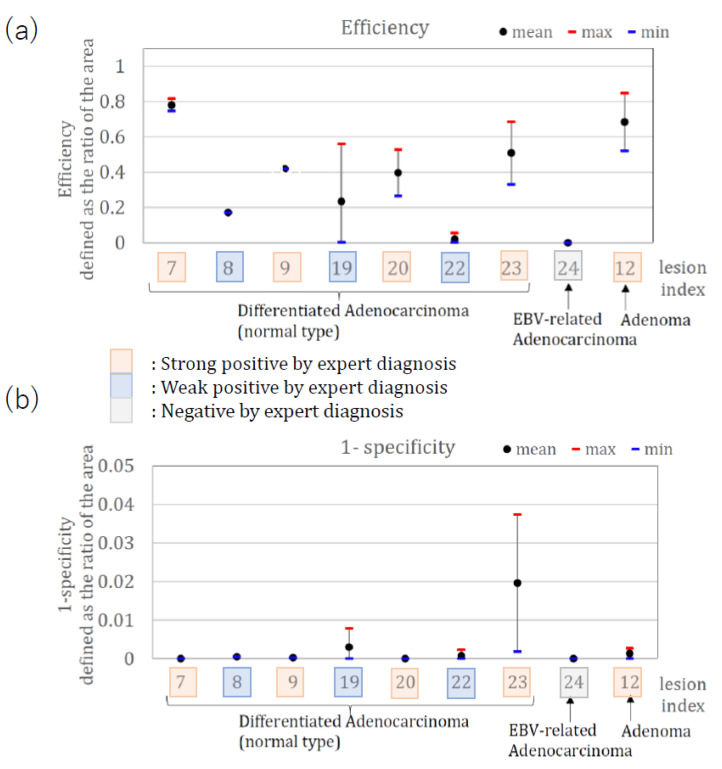
(**a**) Area efficiency (**b**) Area 1-specificity, when method one is applied.

**Figure 5 jcm-11-03030-f005:**
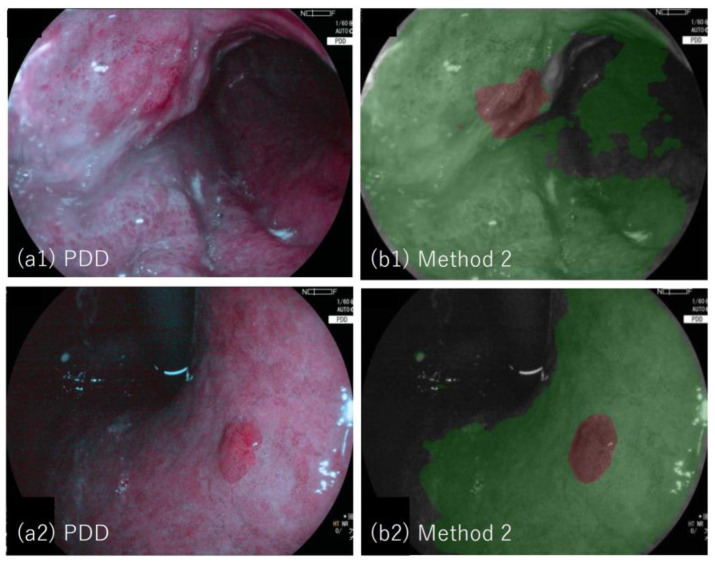
Example of PDD images and the results applied by method two: (**a1**) Photodynamic diagnosis image of lesion #19 (adenocarcinoma, validated lesion), (**b1**) the result of method two overlaid to (**a1**), in which tumor regions determined by method two are depicted in brown and non-tumor regions are depicted in green; (**a2**) photodynamic diagnosis image of lesion #20 (adenocarcinoma, validated lesion),(**b2**) the result of the method two overlaid to (**a2**), in which tumor regions determined by the method two are depicted in brown and non-tumor regions are depicted in green.

**Figure 6 jcm-11-03030-f006:**
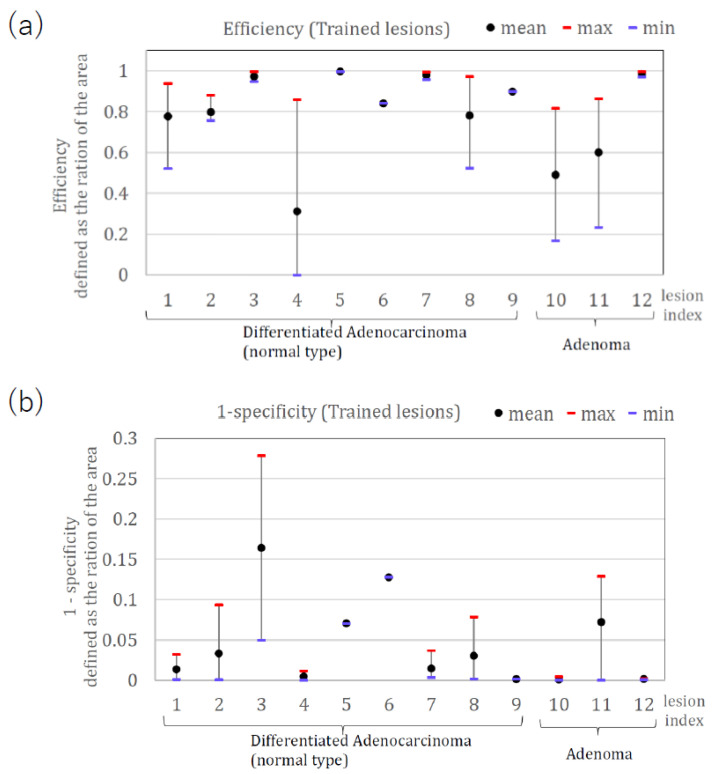
(**a**) Area efficiency (**b**) Area 1-specificity for trained 12 lesions using method two.

**Figure 7 jcm-11-03030-f007:**
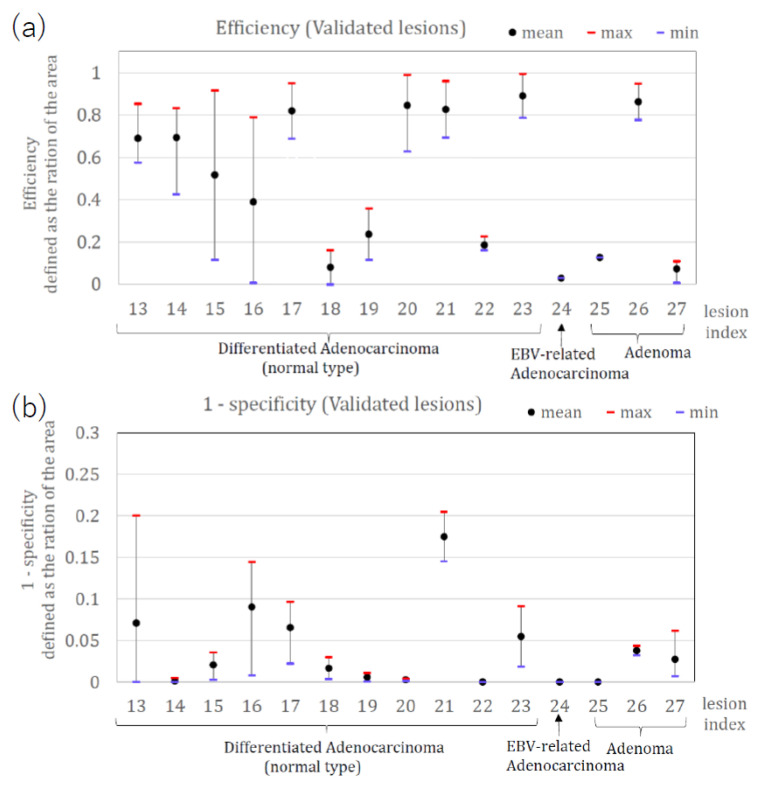
(**a**) Area efficiency (**b**) Area 1-specificity for the validated 15 lesions using method two.

**Figure 8 jcm-11-03030-f008:**
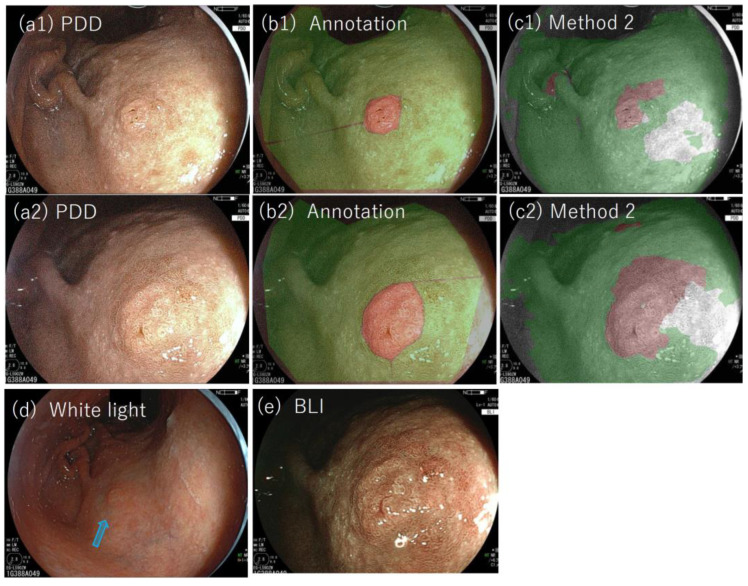
(**a1**,**a2**) Photodynamic diagnosis images of lesion #16 (validated lesion), which was detected as a tumor for the first time on photodynamic diagnosis endoscopy; (**b1**) annotated picture overlaid on (**a1**); (**c1**) the result of method two overlaid on (**a1**); (**b2**) annotated image overlaid on (**a2**); (**c2**) the result of method two overlaid on (**a2**); (**d**) white light image of the lesion in which the tumor location is indicated by a blue arrow; (**e**) blue light image of the lesion.

**Figure 9 jcm-11-03030-f009:**
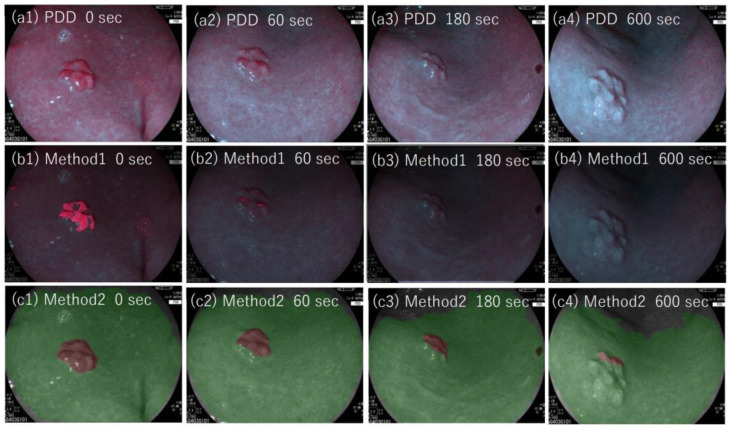
(**a1**–**a4**) Photodynamic diagnosis images of lesion #12 (adenoma), with the photodynamic diagnosis examination times of 0, 60, 180, and 600 s, respectively. Images (**b1**–**b4**) show the results of method one overlaid on (**a1**–**a4**), respectively, in which tumor regions identified by method one are depicted in red. Images (**c1**–**c4**) show results of method two overlaid on (**a1**–**a4**), respectively, in which tumor regions determined by method two are depicted in brown, and non-tumor regions are depicted in green.

**Figure 10 jcm-11-03030-f010:**
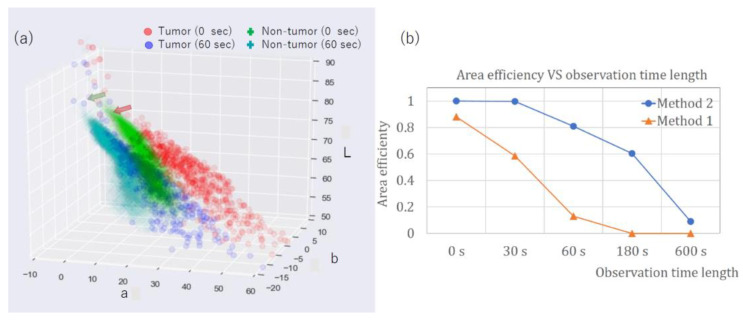
Effect of photobleaching on the area efficiencies: (**a**) Points in the photodynamic diagnosis image of lesion #12 are plotted with corresponding values in LAB color space, for tumor points with observation times of 0 s (red circle) and 60 s (blue circle) and non-tumor points with observation times of 0 s (green cross) and 60 s (cyan cross). Arrows depict the direction of the change from points at 0 s to those at 60 s. (**b**) Area efficiencies of lesion #12 (adenoma), method one (blue circle), and method two (orange triangle), as a function of observation time length.

**Table 1 jcm-11-03030-t001:** Summary of the (**a**) morphology, (**b1**) histological diagnosis before resection (biopsy), (**b2**) histological diagnosis after resection (ESD or surgery), and (**c**) histological depth, of the 27 lesions included in the analysis.

**(a)**							
Adenoma			Adenocarcinoma				
IIa	IIa+IIb	IIc	0-IIa	0-IIb	0-IIc	0-IIa+IIc	0-I
4 (14.8%)	1 (3.7%)	1 (3.7%)	7 (25.9%)	3 (11.1%)	8 (29.6%)	1 (3.7%)	2 (7.4%)
**(b1)**							
Adenoma		Adenocarcinoma					
6 (22.2%)		group4	tub1	tub2	tub1+tub2	tub1+tub2+por	EBV related
		1 (3.7%)	16 (59.3%)	1 (3.7%)	2 (7.4%)	0 (0%)	1 (3.7%)
**(b2)**							
Adenoma		Adenocarcinoma					
6 (22.2%)		tub1		tub2	tub1+tub2	tub1+tub2+por	EBV related
		13 (48.1%)		0 (0%)	5 (18.5%)	2 (7.4%)	1 (3.7%)
**(c)**							
Adenoma		Adenocarcinoma					
M		M		SM1		SM2	
6 (22.2%)		17 (63.0%)		1 (3.7%)		3 (11.1%)	

## Data Availability

Not applicable.

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
