# Peer review of "Objective Methods of 5-Aminolevulinic Acid-Based Endoscopic Photodynamic Diagnosis Using Artificial Intelligence for Identification of Gastric Tumors"

_jcm, 2022, doi:10.3390/jcm11113030_

Round 1
Reviewer 1 Report
This study presents interesting results evaluating to the endoscopic photodynamic diagnosis for gastric tumor identification with 5-aminolevulinic acid. The authors analyzed the results using two main methods with a multineural network and convoluted neural network method. I would like to congratulate the authors for successfully carrying out the arduous research process, including the experiments. I think that the authors spent a lot of effort and time on this study. However, this study was conducted with a small number of enrolled cases, and I consider this study to be a pilot study. In addition, as a study examining the possibility of PDD in differentiated gastric cancer, it is necessary to consider whether it will be useful in actual clinical practice. In other words, it is necessary to consider whether PDD can substitute the existing method and whether the assist role is the main one. The readers will be hoping for more information on this. I have the following comments, which may be helped to increase the article's quality.
Major
- In the introduction section, how about introducing your previous research results (Ann Transl Med. 2020 Mar; 8(5): 178.)? It is easier for readers to understand that this reference is introduced directly in the introduction section rather than appearing in the method section for the first time.
- In the Method section, the researchers should be stated both the IRB number and ethical statement.
- Please explain the ESD procedure briefly in the method section. Most readers did not know this procedure exactly (ESD goal and process).
- In the Result section, there was no information on baseline characteristics of the tumor lesion. Please provide detailed information about the tumor and patient.
- One of the most critical limitations of this study is that the experiment was conducted only on differentiated cancer. As noted by the authors, it is known that 5-ALA uptake is decreased in undifferentiated cancer. If so, I wonder how much PDD will actually help clinical practice in cases where diff/undifferentiated cancer cannot be distinguished when an endoscopist observes the lesion with naked eyes. I recommend that the authors would like to discuss this in the discussion section.
- Moreover, in the case of undifferentiated cancer, especially signet ring cell carcinoma, there are many cases where it cannot be observed conspicuously on the mucosal surface. The role of PDD in these lesions would be more critical, but it is a weakness of this study that the experiment was conducted only in differentiated cancer, which is relatively easy to distinguish from normal mucosa with the naked eye (or with enhanced imaging endoscopy).
- Authors should provide pre-ESD pathology (biopsy sampling) and post ESD pathology (final pathology). Authors should inform the readers about the discrepancy between the two results (pre-ESD pathology and post-ESD pathology) and how accurate the PDD is in the discrepancy cases.
- From the figures, it was estimated that the lesion was at a very early stage (early gastric cancer or adenoma). In the case of early gastric cancer, the entire lesion is not cancer, but adenocarcinoma is identified in some of the lesions originating from adenoma. Is the PDD detection only for the cancer portion? If accompanied by an adenoma portion, can it be distinguished by PDD?
- To confirm the above issue, the experiment on advanced gastric cancer may be helpful. It is presumed that the authors performed the experiment in EGC where the lesion is less clear than in AGC that can be visually distinguished, but I think that it will be helpful for validation as a diagnostic tool of PDD to perform validation in AGC first and then proceed in EGC. Please consider this issue in the following study.
- The authors noted that 4 out of 29 lesions were identified for the first time in PDD. In 13.8% of cases, the lesion was missed in the endoscopy. Isn't the endoscope accuracy too low?
- The contents of the Discussion section were poor, and the description of the study limitation is necessary based on the above comments. Authors should allocate and organize the Limitation section separately.
Author Response
First of all, thank you very much for reviewing of my article.
・We have incorporated comments from reviewers. We have found inappropriate point with regard to the selection of the lesions. 3 lesions in 29 lesions (#27, #28, #29) were not resected until now and followed up regularly (because they are adenoma of small size). The 3 lesions are removed from the analysis.
・One lesion(#9) which should be included, was not, which is now included in the analysis. Therefore, total analyzed lesions are 27.
We apologize for those. It is correct now.
- In the introduction section, how about introducing your previous research results (Ann Transl Med. 2020 Mar; 8(5): 178.)? It is easier for readers to understand that this reference is introduced directly in the introduction section rather than appearing in the method section for the first time.
We have included our results on introduction section.
2. In the Method section, the researchers should be stated both the IRB number and ethical statement.
We included IRB number and ethical statement in the last part of the article, following the template of MDPI:
Institutional Review Board Statement: The study was conducted in accordance with the Declaration of Helsinki, and approved by the Institutional Ethics Committee of Tottori University (protocol code CRB6200003).
However if the position should be at the method section, I will change following the instruction.
3. Please explain the ESD procedure briefly in the method section. Most readers did not know this procedure exactly (ESD goal and process).
I have included the procedure in the method section:
ESD was performed after the PDD screening, according to the standard procedure [15]: marking, injecting liquid composed of hyaluronic acid and saline into submucosa, making incision and dissecting with submucosa level. Light shading of 24 hours after 5-ALA intake was implemented to avoid any photosensitivity [7].
4. In the Result section, there was no information on baseline characteristics of the tumor lesion. Please provide detailed information about the tumor and patient.
We have now include the characteristics of the lesions in the method section. We have also included patient characteristics:
27 lesions in 20 patients (age from 58 to 78 (median 73), two are female) of differentiated adenocarcinoma and adenoma were included in this study.
5. One of the most critical limitations of this study is that the experiment was conducted only on differentiated cancer. As noted by the authors, it is known that 5-ALA uptake is decreased in undifferentiated cancer. If so, I wonder how much PDD will actually help clinical practice in cases where diff/undifferentiated cancer cannot be distinguished when an endoscopist observes the lesion with naked eyes. I recommend that the authors would like to discuss this in the discussion section.
We have included the PDD role and limitation in the discussion and limitation section, following your comments.
6. Moreover, in the case of undifferentiated cancer, especially signet ring cell carcinoma, there are many cases where it cannot be observed conspicuously on the mucosal surface. The role of PDD in these lesions would be more critical, but it is a weakness of this study that the experiment was conducted only in differentiated cancer, which is relatively easy to distinguish from normal mucosa with the naked eye (or with enhanced imaging endoscopy).
The low efficiency due to the non-tumor gland at the surface blunts the 5-ALA based PDD. This is significant limitation and noted on the discussion and limitation section. To mitigate this, it can be considered that green light is used in the PDD, not blue light currently used. However it can lower light intensity significantly.
7. Authors should provide pre-ESD pathology (biopsy sampling) and post ESD pathology (final pathology). Authors should inform the readers about the discrepancy between the two results (pre-ESD pathology and post-ESD pathology) and how accurate the PDD is in the discrepancy cases.
We have provided pre and post resected pathology in the Table 1. There are three lesions which have discrepancy. We do not have difference between adenoma and adenocarcinoma for 5-ALA based PDD untill now.
8. From the figures, it was estimated that the lesion was at a very early stage (early gastric cancer or adenoma). In the case of early gastric cancer, the entire lesion is not cancer, but adenocarcinoma is identified in some of the lesions originating from adenoma. Is the PDD detection only for the cancer portion? If accompanied by an adenoma portion, can it be distinguished by PDD?
We have not seen difference between adenoma and differentiated adenocarcinoma for the efficiency of PDD, although with the small number of cases.
9. To confirm the above issue, the experiment on advanced gastric cancer may be helpful. It is presumed that the authors performed the experiment in EGC where the lesion is less clear than in AGC that can be visually distinguished, but I think that it will be helpful for validation as a diagnostic tool of PDD to perform validation in AGC first and then proceed in EGC. Please consider this issue in the following study.
We had only 4 lesions of SM invasions in the analysis. Three lesions had good efficiency for 5-ALA PDD, one lesion of EBV-related carcinoma was low efficiency. As you mentioned, It is good to apply 5-ALA PDD to AGC to assess the difference of efficiency for various component of the histology. We will have included AGC samples in the following 5-ALA PDD research.
10. The authors noted that 4 out of 29 lesions were identified for the first time in PDD. In 13.8% of cases, the lesion was missed in the endoscopy. Isn't the endoscope accuracy too low?
One case among the first identified lesions was in the three cases we have removed from the analysis. So now 3/27 lesions were first detected by PDD. The frequency was still high, but the morphology of the lesions were IIb or small IIc, which would have missed, I would think.
11. The contents of the Discussion section were poor, and the description of the study limitation is necessary based on the above comments. Authors should allocate and organize the Limitation section separately.
We have separated discussion section and limitation section, and add comments according to your comments.
Could you please re-review the updated version of the article,
Best regards and thank you very much in advance,
Taro Yamashita

Reviewer 2 Report
A very good article with high novelty; however, still premature.
Iam not sure what are the numbers you gave for the tumor cases!!?? and I prefer you give their diagnosis; it is very confusing.
The value of your methods has to be illustrated as regard to highly differentiated adenocarcinoma; whether can be applied or not??
The cost has to be evaluated of both methods as it can be an important factor for application of either of them.
What are the indications to apply the PDD methods in real life??
It is not clear if you can apply this methods for hyperplastic lesions or GIST tumors? Also, Is presence of other associated lesions can interfere with the application and efficacy of PDD??
Author Response
First of all, thank you very much for reviewing of my article.
・We have incorporated all the comments from reviewers. We have found inappropriate point with regard to the selection of the lesions. 3 lesions in 29 lesions (#27, #28, #29) were not resected until now and followed up regularly (because they are adenoma of small size). The 3 lesions are removed from the analysis.
・One lesion(#9) which should be included, was not, which is now included in the analysis. Therefore, total analyzed lesions are 27.
We apologize for those. It is correct now.
>I am not sure what are the numbers you gave for the tumor cases!!?? and I prefer you give their diagnosis; it is very confusing.
We prepared table (Table 1) summarizing characteristics of the lesions.
>The value of your methods has to be illustrated as regard to highly differentiated adenocarcinoma; whether can be applied or not??
I am very sorry, I cannot understand full meaning of your comments due to my English ability. Developed methods had been applied to the differentiated carcinoma up to now, since 5-ALA based PDD do not have much sensitivity for pure poorly differentiated carcinoma. This point is the limitation of this study and 5-ALA based PDD itself. We mentioned in the limitation section.
>The cost has to be evaluated of both methods as it can be an important factor for application of either of them.
We have included comments on cost on discussion section:
"With the achieved objectivity and ability, the developed methods could make PDD endoscopy to be valuable for general screening of upper gastrointestinal tract. Currently, the application is limited to the occasion of detailed inspection, due to the cost of 5-ALA (~ 600 dollars per one PDD endoscopy) and the need of hospitalization for 24-hours light shading."
We have also included indication for processing time (time cost) in the limitation section:
”The procedures were developed on Python software platform and took from several to ten seconds per one image executing on personal computer. For real time application during endoscopic examination, the processing time should be shortened, which is possible by executing on C++ software platform or with the specialized system of small latency, for example using real-time processor.”
>What are the indications to apply the PDD methods in real life??
It is not clear if you can apply this methods for hyperplastic lesions or GIST tumors? Also, Is presence of other associated lesions can interfere with the application and efficacy of PDD??:
We think PDD is valuable tool for detecting tumor without missing. However the cost is not low and we need hospitalization in order to avoid photosensitivity, which limits PDD application. We have included comments on the PDD role in the endoscopy in the discussion section:
”With the achieved objectivity and ability, the developed methods could make PDD endoscopy to be valuable for general screening of upper gastrointestinal tract. Currently, the application is limited to the occasion of detailed inspection, due to the cost of 5-ALA (~ 600 dollars per one PDD endoscopy) and the need of hospitalization for 24-hours light shading.”
Best regards and thank you very much in advance,
Taro Yamashita

Reviewer 3 Report
Here are my comments to improve the manuscript:
1.) Introduction should be rewritten to reflect the importance of this research.
2.) Are there any similar methods available or reported?
3). When selecting images for analysis, how did you make sure that there was/or no significant difference between the images? Is any statistical analysis done to see if there are any differences?
4).How did you calibrated the methods?
Overall, the paper requires major editing.
Author Response
First of all, thank you very much for reviewing of my article.
・We have incorporated all the comments from reviewers. We have found inappropriate point with regard to the selection of the lesions. 3 lesions in 29 lesions (#27, #28, #29) were not resected until now and followed up regularly (because they are adenoma of small size). The 3 lesions are removed from the analysis.
・One lesion(#9) which should be included, was not, which is now included in the analysis. Therefore, total analyzed lesions are 27.
We apologize for those. It is correct now.
1.) Introduction should be rewritten to reflect the importance of this research.
We have rewritten the introduction section, commenting the current situation of applying artificial intelligence to endoscopy, and with focus on the importance of this research.
2.) Are there any similar methods available or reported?
There are several reports applying artificial intelligence to the endoscopy. We have comments in the introduction section:
"Applying artificial intelligence (AI), especially CNN method, to white-light or NBI endoscopy has been recently investigated [10], where AI applications can have more accurate performance of detecting abnormality than endoscopists. With the CNN method, not only identifying objects but also segmenting images into classified objects can be performed [8]. Performing segmentation according to the estimated histology with CNN method has recently been investigated for colonic endoscopy [11, 12]. For gastric endoscopy, we have not seen any systematic study of performing segmentation with artificial intelligence. "
3). When selecting images for analysis, how did you make sure that there was/or no significant difference between the images? Is any statistical analysis done to see if there are any differences?
For majority of the lesions(22 in 27 lesions), we have included several images possibly with different viewpoints for each lesion, with which we think it is possible to estimate the uncertainty coming from the selection of the images. In the result figure of efficiency and specificity, it is shown as error bar, with the center denoting the average.
4).How did you calibrated the methods?
We have tuned learning procedure for both method one and two. We have included comments on that:
For method one:
"The parameter of learning procedure, for example, node size of each layer of neural network (= 300, 200 and 100, for the first, second and third layer, respectively) or activation function (= logistic) were tuned or selected, such that the learning converged faster with smaller error."
For method two:
"Lesions were divided such that both adenoma and adenocarcinoma lesions were included in training and validation. Number of epochs for CNN training, which was the number of multiple usage of data sets, was selected as 120, considering that error calculated for validated lesions stopped improving at around 120 epochs."
We have re-examined the title of the article. We would think it could be better to include the word "artificial intelligence" in the title:
"Objective methods of 5-aminolevulinic acid-based endoscopic photodynamic diagnosis using artificial intelligence for identification of gastric tumors"
Could you please re-review the updated version of the article,
Best regards and thank you very much in advance,
Taro Yamashita

Round 2
Reviewer 1 Report
The authors reflected most of my opinions. Thank you for your efforts. However, the baseline characteristics information for tumor lesions should not be summarized as in Table 1. Table 1 should be moved to supplement materials. A table showing baseline characteristics should consist of the number, percent, and mean (or median) values corresponding to each variable. The table should contain the arithmetic statistics data. I propose that the authors re-create table 1 and move the existing table 1 to supplementary materials. In addition, the new table 1 for baseline characteristics must be located in the Result section.
Author Response
Thank you very much for your reviewing the article.
We have provided long diameter and estimated area of lesions, and created tables summarizing morphology, histological diagnosis and histological depth of 27 lesions, following your suggestion :
The longest diameter of the lesion included in the analysis was from 5 to 40 mm (median 18 mm), and estimated area, assuming the lesion was the ellipse, was from 11.8 mm2 to 1068.1 mm2 (median 181.4 mm2). Table 1(a)(b)(c) show the summary of the morphology, histological diagnosis and histological depth, respectively, of the 27 lesions included in the analysis.
Best regards,
Taro Yamashita

Reviewer 3 Report
The authors have made the necessary changes to the manuscript and it can now be accepted for publication.
Author Response
Thank you very much for your reviewing our article.
We are very grateful your valuable comments.
At the latest version, we have moved Table 1 to supplementary section according to the instruction of other reviewer, and created tables summarizing features of the lesions:
The longest diameter of the lesion included in the analysis was from 5 to 40 mm (median 18 mm), and estimated area, assuming the lesion was the ellipse, was from 11.8 mm2 to 1068.1 mm2 (median 181.4 mm2). Table 1(a)(b)(c) show the summary of the morphology, histological diagnosis and histological depth, respectively, of the 27 lesions included in the analysis. Further detailed features of the lesions are shown in Supplementary Table 1.
Best regards,
Taro Yamashita